# Free roaming of 3D stratum models based on internal and external boundary identification

**Yusen Zhong**[1,2], **Zhen Liu**[1,2]*, **Cuiying Zhou**[1,2]

**1** School of Civil Engineering, Sun Yat-sen University, Guangzhou, P. R. China, **2** Guangdong Engineering Research Centre for Major Infrastructure Safety, Guangzhou, P. R. China

* liuzh8@mail.sysu.edu.cn

**Data Availability Statement:** All relevant data are within the manuscript and its Supporting Information files.

**Funding:** Thank you for reminding me. We have stated in the cover letter as follows" This research

## Abstract

3D stratum roaming can visualize the complex geomorphology, underground structure and stratum distribution of geotechnical engineering. Conventional 3D stratum roaming technology is generally aimed at roaming outside the 3D stratum model but rarely roams inside the 3D stratum model, and the efficiency of switching between external and internal roaming is low. In practical engineering, especially in geological and geotechnical engineering, the underground structure and stratum situation are critical. Therefore, focusing on this problem, this paper adopts a three-dimensional roaming engine to connect the inside and outside of a three-dimensional model. Based on an internal and external boundary identification method, the combination of external roaming and internal roaming of the three-dimensional stratum model is implemented by using a stratum virtual surface, and lightweight loading is carried out by controlling the stratum virtual surface to establish the internal and external roaming of the 3D stratum model, and to give full play to the advantages of lightweight loading to provide more intuitive and comprehensive geological information for the project.

## 1 Introduction

3D underground roaming technology is of great importance in geological and geotechnical engineering. 3D underground roaming provides a better understanding of underground regions. In geological and geotechnical engineering, underground construction is often more complex and dangerous than surface construction [1]. 3D underground roaming technology can be used to simulate and plan underground construction in a virtual environment, eliminating all possible problems in advance, thereby reducing construction risks, improving construction efficiency and reducing construction costs. In addition, 3D underground roaming technology can present underground information more intuitively, roaming in the underground virtual environment to accurately predict and assess the feasibility and safety of underground projects. The internal roaming of the 3D stratum model based on internal and external recognition indirectly establishes 3D underground roaming, which enables the 3D stratum model to intuitively display the geological structure, spatial topology and lithology of the strata, providing accurate information for the analysis of geological structures, fault distribution and

# PLOS ONE

is supported by the National Natural Science Foundation of China (Grant Numbers: 42293354, 42293351, 42293355, 42277131, 41977230). These fundings are all awarded by the corresponding author Cuiying Zhou. We declare that the funder has no known competing financial interests or personal relationships that could have appeared to influence the work reported in this paper. The funders had no role in study design, data collection and analysis, decision to publish, or preparation of the manuscript.".

**Competing interests:** The authors have declared that no competing interests exist.

other work, as well as providing a reliable basis for the development and utilization of underground space.

Three-dimensional roaming [2–4] was previously presented as two-dimensional maps and panoramic photographs [5–7]. The advent of 3D models has given 3D roaming improved realism and a higher sense of three-dimensionality. To obtain better 3D results, many scholars [8–18] have overcome the limitations of the traditional roaming techniques, which were only applicable to 3D scenes with small data volumes and simple models. This was done by rebuilding models, improving model accuracy and using the best roaming engines. At present, a three-dimensional model [19] is usually constructed by a set of mosaic surfaces [20], that is, a three-dimensional stratum model is a surface model, so the three-dimensional roaming technology is often roaming the outside of the three-dimensional model. One of the surface model's most important features is its hollowness. When the viewpoint moves to the model's interior, it will appear hollow and cannot simulate the actual situation. At the same time, conventional 3D scene roaming does not consider the viewpoint of the model's interior. Although Zhang [21] uses VRML and 3DSMAX as the development platform to implement virtual roaming of an interior of a residential building in a residential community, the interior of the scene is still outside the model, and the viewpoint does not enter the interior of the model; if the roaming viewpoint enters the interior of the model, it will not reflect the real information inside the model. There are also studies that have been carried out to reduce the influence hollow models by making the stratum model transparent to approximate the internal roaming of the 3D stratum model. Nevertheless, in practice, this method has significant limitations. In addition to the internal roaming of the model, it is also important to switch between external and internal roaming of the stratum model. By switching between the 3D underground roaming scenes, the spatial distribution and relationships of the underground structures can be better understood and presented more intuitively for better design and construction. Among others, Tao's [22] WebGL-based virtual roaming method achieves a seamless connection between video and panorama. Zhou [23] proposed a new dual-mode 3D terrain landscape construction and display method that implements dynamic roaming and seamless switching of two 3D scenes but requires reloading the scene for scene switching. The FRANK S [24] A combination of 3D internal and external roaming is achieved through a 3D mapping component and a 3D walkthrough component, where the 3D mapping component is an external model created by a 3D modeling program and the 3D walkthrough component is an internal model created by a photo stitching program. Some scholars have also used two trackers to position the camera and thus switch between 3D internal and external roaming, however, this switching relies on two trackers and is relatively inefficient. In summary, the current implementation of 3D stratum internal roaming relies on the transparency of the model to weaken the influence of the hollow model to enhance the visual effect, which cannot truly solve the problem of hollow model internal roaming, and the switching between internal and external roaming is very complicated, leading to low efficiency. In the field of geology and geotechnical engineering, the internal structure of geological bodies is often the most important, and whether the detailed preview of the internal structure of geological bodies can be realized is of great significance in practical engineering.

To this end, this paper is based on traditional 3D model external roaming, which is based on the 3D roaming drawing protocol (WebGL) and the 3D roaming engine (Three.js) to achieve free roaming inside the 3D stratum by creating a series of stratum virtual surfaces for visual deception. Through the inner and outer boundary recognition method, the model's external roaming and internal roaming can be achieved in the same scene without scene switching. Among them, the 3D roaming drawing protocol is a cross-platform JavaScript interface for rendering 3D graphics in the browser without the need for plug-ins, and it is

available for all major browsers [25]. The 3D roaming engine is an open-source framework for the 3D roaming drawing protocol, allowing complex 3D scenes to be created with a small amount of JavaScript code [26]. Ultimately, the combination of external and internal roaming of the 3D models is achieved using internal and external boundary recognition, stratum virtual surfaces and the control processing of the stratum virtual surfaces.

In this paper, we first introduce the mechanism of stratum virtual plane, and realize the inner roaming of 3D stratum through stratum virtual plane. Secondly, the internal and external boundary recognition method is introduced, which can realize the quick switching between 3D stratum internal roaming and 3D stratum external roaming. Finally, the feasibility study of this technology is carried out through a case project.

## 2 Research content and methodology

This paper first introduces, in detail, the formation mechanism of the stratum virtual surfaces, and the introduction of stratum virtual surfaces in the 3D stratum model to obtain an internal roaming model based on the internal roaming model. This is combined with the external roaming model for a specific analysis of the combination of internal and external roaming through internal and external identification. The particular research content is shown in Fig 1.

### 2.1 Internal roaming and internal and external boundary identification method for 3D stratum models

The 3D stratum model is a surface model with a hollow interior. When the camera is moved to the interior of the model, it appears that the interior of the model is hollow, and it is impossible to observe the specifics of the interior of the model. In this paper, we need to create a stratum virtual surface to simulate the real distribution of the interior of the model to "cheat" the field

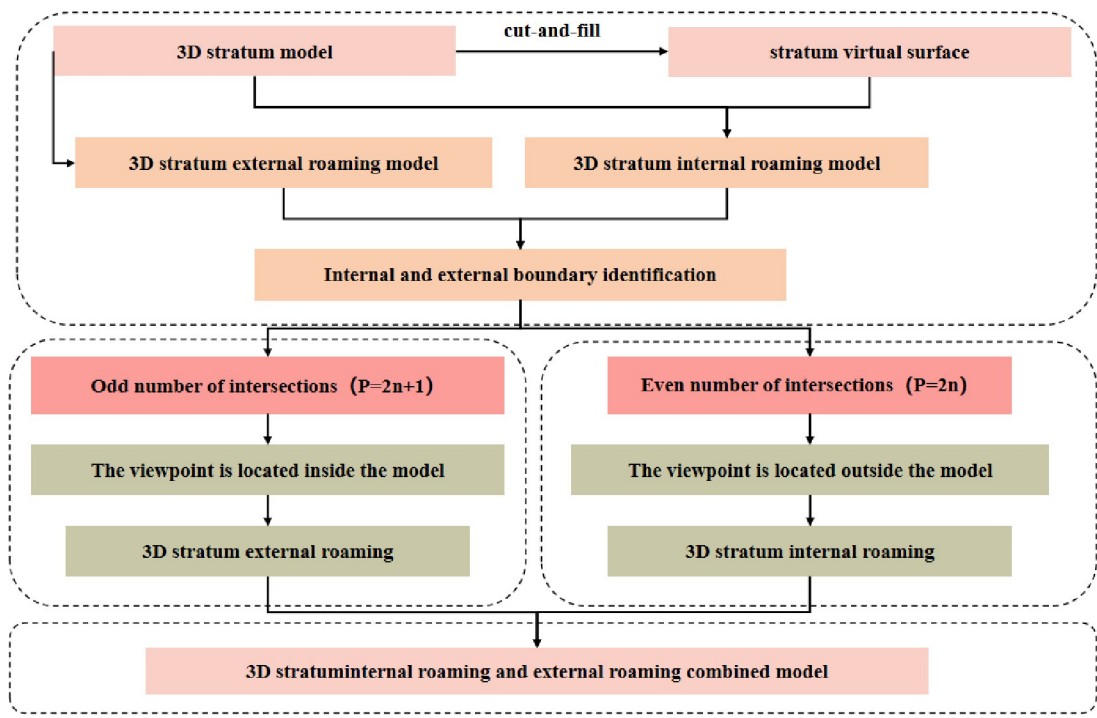

**Fig 1. Schematic diagram of the study.**

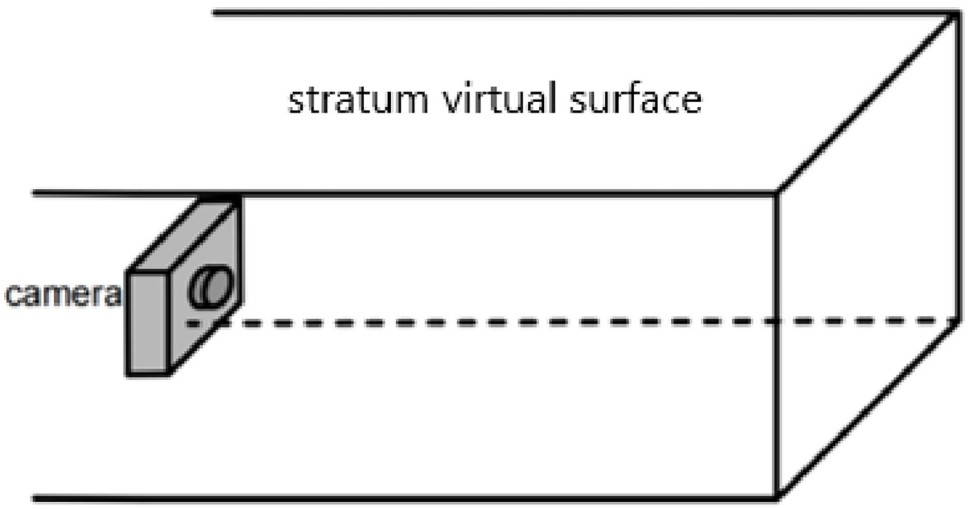

**Fig 2. Schematic diagram of the stratum virtual surface.**

of view. As shown in Fig 2, five stratum virtual surfaces are created based on the distance between the front, left, right, above and below the location of the internal roaming camera (viewpoint), and the actual situation inside the model is loaded on the five stratum virtual surfaces.

After implementing the internal roaming of the 3D stratum model, external roaming is combined with internal roaming by means of internal and external boundary recognition methods. The internal and external identification method uses the camera as the source point, creates internal and external identification rays in the direction of the line of sight, and divides the internal and external roaming by the number of intersections between the internal and external identification rays and the 3D stratum model.

## 2.2 Formation mechanisms for the stratum virtual surfaces and control of the clipping surface

The stratum virtual surface is loaded with a lithological texture pattern, and the clipping surface cut-and-fill idea is used, giving the five stratum virtual surfaces a lithological texture pattern through the clipping surface, and finally allowing the five stratum virtual surfaces with the lithological texture pattern obtained through the clipping surface to follow the internal roaming camera, constantly updating the location of the clipping surface and the lithological pattern. This completes the real-time update of the lithological texture on the stratum virtual surface.

**2.2.1 Formation of stratum virtual surfaces by clipping surface cut-and-fill.** The clipping surface cut-and-fill is based on the stencil test of the stratum cut function and requires two stencil tests, and the principle is shown in Fig 3:

The process is as follows:

1. First Stencil Test. Create a stencil mesh with the geometry of the stratum model to be profiled, turn off color writing, which is invisible and only provides a caching role, and turn on the first stencil test, make

$$S_{St} = 0$$

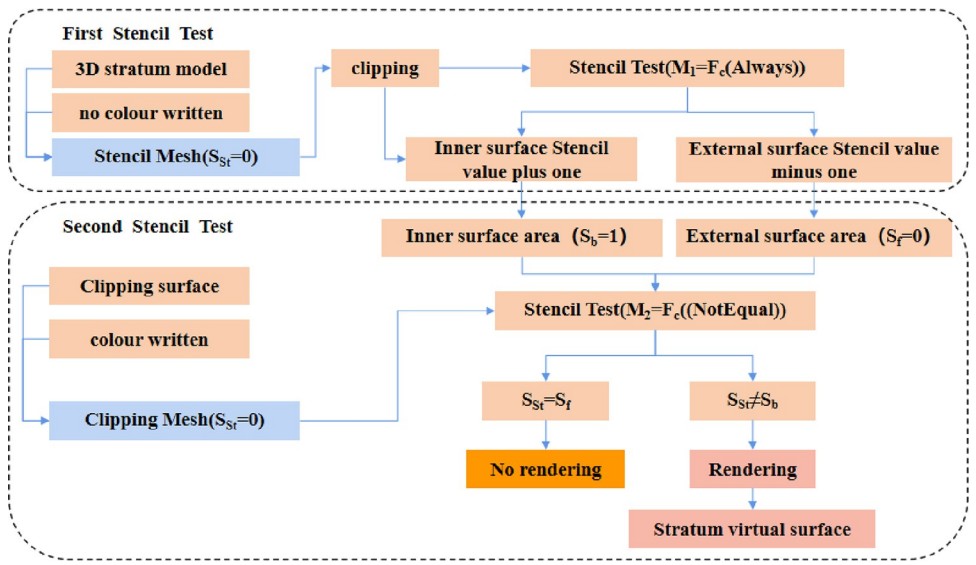

**Fig 3. Principle of the stencil Test of the stratum cut.**

$$M_1 = F_c(\text{Always})$$

$$P = \left\{ \begin{array}{l} X_1(1,0,0) \\ X_2(-1,0,0) \\ Y(0,1,0) \\ Z_1(0,0,1) \\ Z_2(0,0,-1) \end{array} \right\}$$

where $S_{St}$ is the default Stencil value for the stencil mesh (stencil values range from 0 to 255), $M_1$ is the first stencil test comparison function, and $F_c$ (always) means that the test is passed regardless of the stencil value. P is the set of cutting planes. $X_1$, $X_2$, Y, $Z_1$ and $Z_2$ are the five profiling planes, $X_1$ (1,0,0) represents the profiling plane with (1,0,0) as the direction vector, and the stencil mesh materials with a negative distance from it will not be rendered.

2. After completing the first stencil test, the stencil mesh exposes the inner surface of the model due to clipping, as shown in Figs 4 and 5, and the outer surface stencil value S of the stencil mesh is subtracted by one, and the inner surface stencil value is incremented by one.

At this time:

$$S_A = S_{St} - 1$$

$$S_B = S_{St} + 1$$

$$S_C = S_{St} - 1$$

where $S_A$ and $S_C$ are the stencil values of area A and area C, which are also the stencil values of

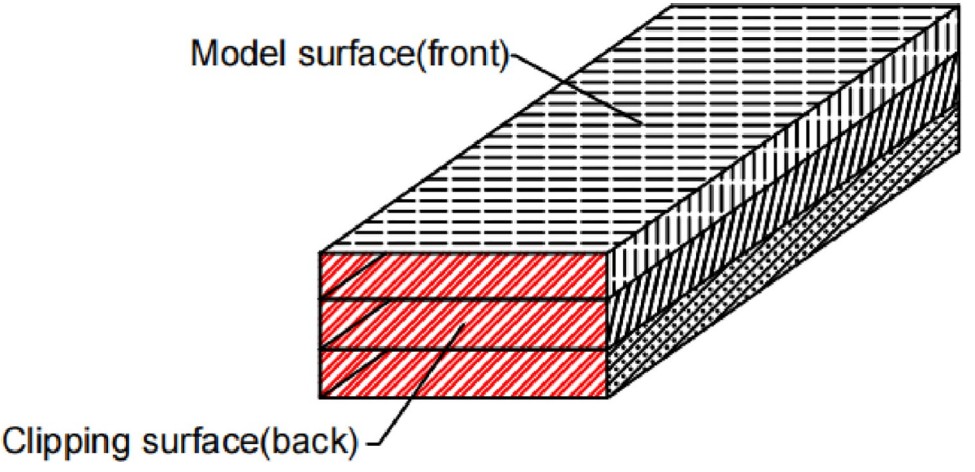

**Fig 4. Schematic diagram of the inner and outer surfaces of the stencil mesh.**

the outer surface of the model; $S_B$ is the stencil value of area B, which is also the stencil value of the inner surface of the model exposed by the cut; the stencil values range from 0 to 255. $S_A = 0$, $S_B = 1$ and $S_C = 0$.

3. After completing the calculation of the stencil values for the internal and external surfaces, the second stencil test is opened. Create the clipping mesh with the geometry model of plane $X_1$, plane $X_2$, plane $Y$, plane $Z_1$, and plane $Z_2$. Make

$$S_{cl} = 0$$

$$S_{ref} = 0$$

$$M_2 = F_c(\text{NotEqual})$$

$S_{cl}$ is the default stencil value for the clipping mesh (the stencil value takes values from 0 to

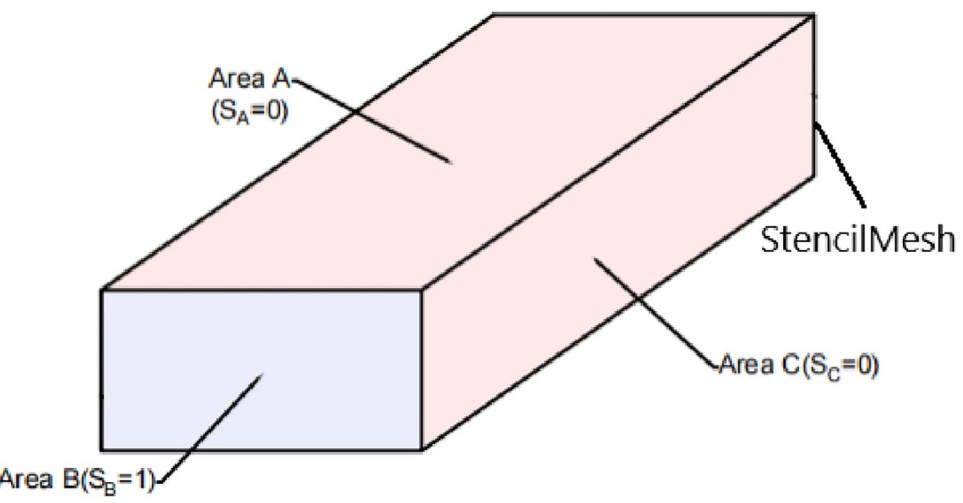

**Fig 5. Schematic diagram of the stencil mesh area.**

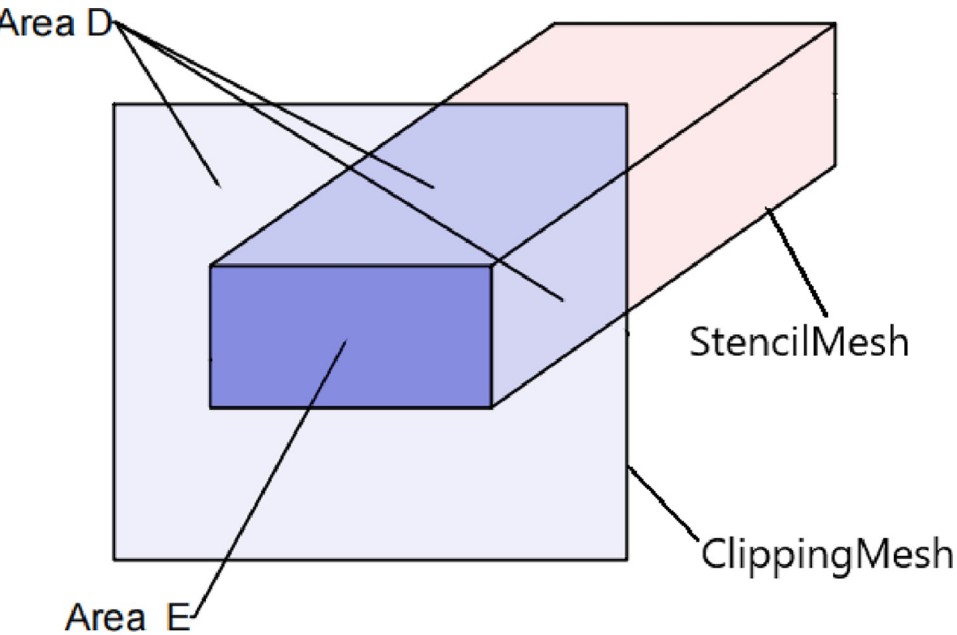

**Fig 6. Schematic diagram of the location of the clipping mesh and the stencil mesh.**

255), $S_{ref}$ is the standard stencil value, $M_2$ is the second stencil test comparison function, and $F_c$ (NotEqual), which means that the test can be passed when the stencil value is not the same as the standard stencil value.

Based on the intersection line between the clipping mesh and the stencil mesh, the clipping mesh was divided into Area D and Area E, as shown in Fig 6. Based on the calculations of the first stencil test, the stencil values $S_D$ and $S_E$ for the clipping mesh were

$$S_D = 0$$

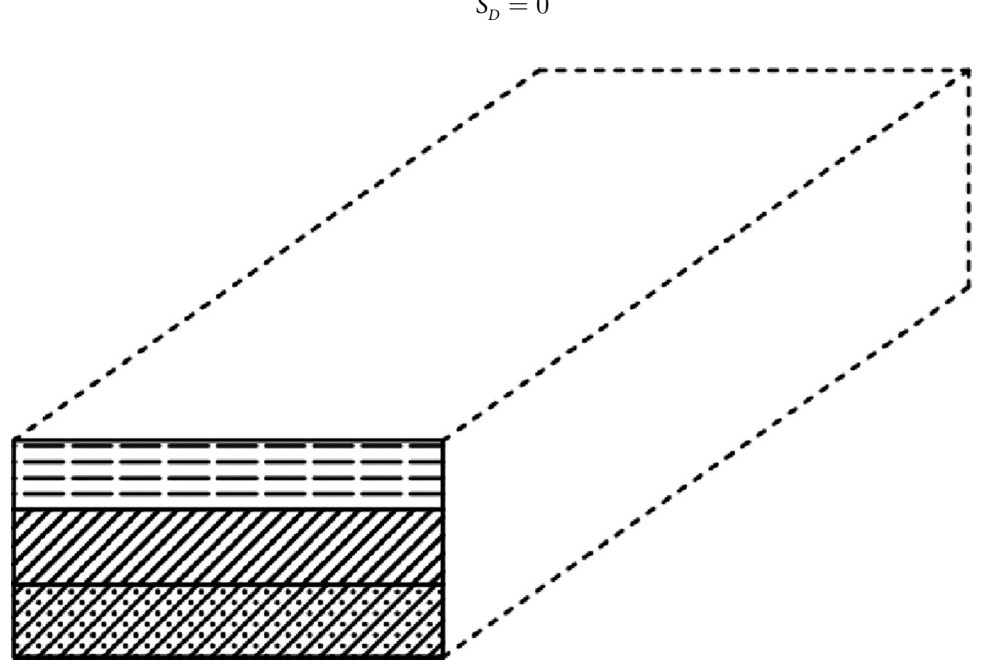

**Fig 7. Schematic diagram of the clipping surface.**

$$S_E = 1$$

$S_D$ is the stencil value of area D, and $S_E$ is the stencil value of area E.

Since $S_D = S_{ref}$, Region D will be rendered if it passes the test; $S_E \neq S_{ref}$, Region E will not be rendered if it does not pass the test, and the stencil mesh is an invisible model. The rendering results are shown in Fig 7.

4. Determine the render order. The clipping mesh should be rendered after the stencil mesh, i.e., after the first template test. Assuming that the stratum model has i layers and the render order is rendered according to the size of the render order value, then the render order value for the stencil mesh is(R is the order value of the stencil mesh):

$$R = \begin{Bmatrix} R_1 \\ R_2 \\ R_3 \\ R_4 \\ \vdots \\ R_i \end{Bmatrix} = \{ X_1 \quad X_2 \quad Y \quad Z_1 \quad Z_2 \}$$

$$R = \begin{Bmatrix} 1 + (5 \times 0) & 2 + (5 \times 0) & 3 + (5 \times 0) & 4 + (5 \times 0) & 5 + (5 \times 0) \\ 1 + (5 \times 1) & 2 + (5 \times 1) & 3 + (5 \times 1) & 4 + (5 \times 1) & 5 + (5 \times 1) \\ 1 + (5 \times 2) & 2 + (5 \times 2) & 3 + (5 \times 2) & 4 + (5 \times 2) & 5 + (5 \times 2) \\ 1 + (5 \times 3) & 2 + (5 \times 3) & 3 + (5 \times 3) & 4 + (5 \times 3) & 5 + (5 \times 3) \\ \vdots & \vdots & \vdots & \vdots & \vdots \\ 1 + (5 \times (i-1)) & 2 + (5 \times (i-1)) & 3 + (5 \times (i-1)) & 4 + (5 \times (i-1)) & 5 + (5 \times (i-1)) \end{Bmatrix}$$

The rendering order values for the clipping mesh are(C is the order value of the clipping mesh):

$$C = \begin{Bmatrix} C_1 \\ C_2 \\ C_3 \\ C_4 \\ \vdots \\ C_i \end{Bmatrix} = \{ X_1 \quad X_2 \quad Y \quad Z_1 \quad Z_2 \}$$

$$C = \begin{Bmatrix} 1.1 + (5 \times 0) & 2.1 + (5 \times 0) & 3.1 + (5 \times 0) & 4.1 + (5 \times 0) & 5.1 + (5 \times 0) \\ 1.1 + (5 \times 1) & 2.1 + (5 \times 1) & 3.1 + (5 \times 1) & 4.1 + (5 \times 1) & 5.1 + (5 \times 1) \\ 1.1 + (5 \times 2) & 2.1 + (5 \times 2) & 3.1 + (5 \times 2) & 4.1 + (5 \times 2) & 5.1 + (5 \times 2) \\ 1.1 + (5 \times 3) & 2.1 + (5 \times 3) & 3.1 + (5 \times 3) & 4.1 + (5 \times 3) & 5.1 + (5 \times 3) \\ \vdots & \vdots & \vdots & \vdots & \vdots \\ 1.1 + (5 \times (i-1)) & 2.1 + (5 \times (i-1)) & 3.1 + (5 \times (i-1)) & 4.1 + (5 \times (i-1)) & 5.1 + (5 \times (i-1)) \end{Bmatrix}$$

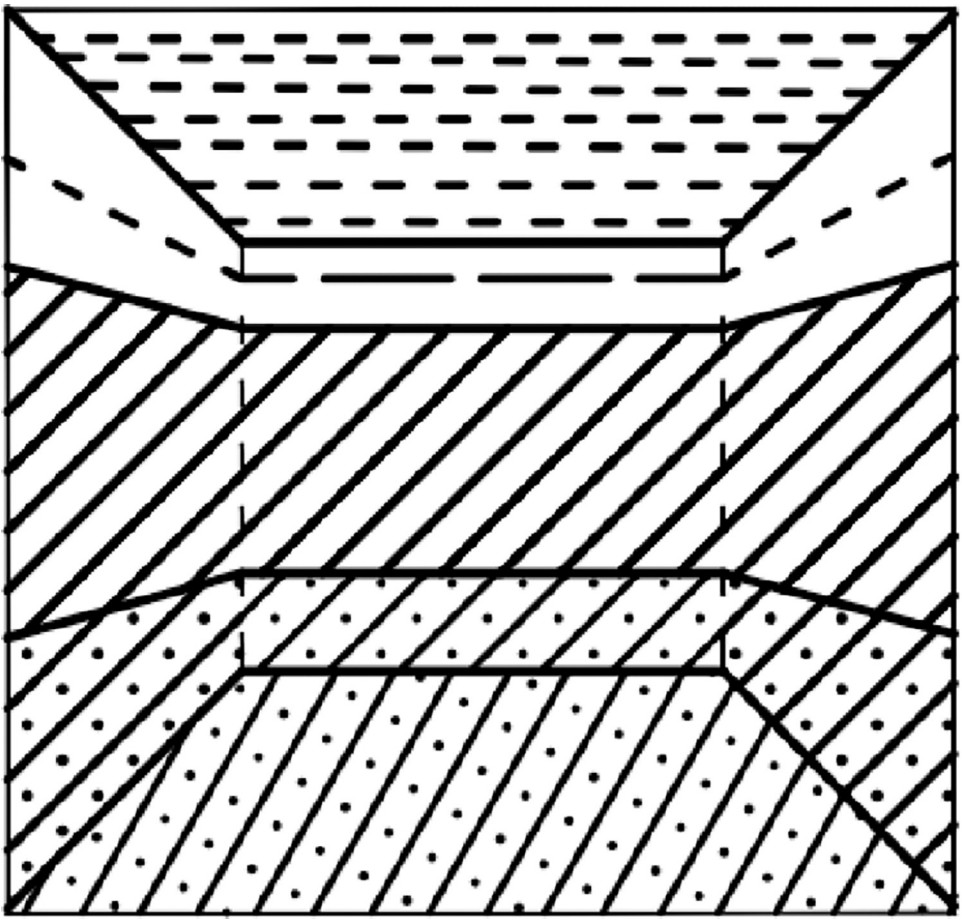

**Fig 8. Rendering of five clipping surfaces.**

The results of the five clipping surfaces rendered in turn are shown in Fig 8:

**2.2.2 Control of the stratum virtual surface.** The stratum virtual surface is the clipping surface. Once the five stratum virtual surfaces are completed, the 3D internal roaming is approximated by controlling the five stratum virtual surfaces to follow the camera's movement. As shown in Fig 9, each stratum virtual surface is an overall clip of the model in one direction. Therefore, when moving, i.e., When the camera position is updated, it is not necessary to update all five stratum virtual surfaces simultaneously, only a few that affect the visual effect of the roaming, achieving lighter loading.

Assume that the distances from the internal roaming camera (viewpoint) to each of the five stratum virtual surfaces $X_1$, $X_2$, $Y$, $Z_1$ and $Z_2$ are(L is the distance set):

$$L = \begin{Bmatrix} X_1 \\ X_2 \\ Y \\ Z_1 \\ Z_2 \end{Bmatrix} = \begin{Bmatrix} l_1 \\ l_1 \\ l_2 \\ l_1 \\ l_1 \end{Bmatrix}$$

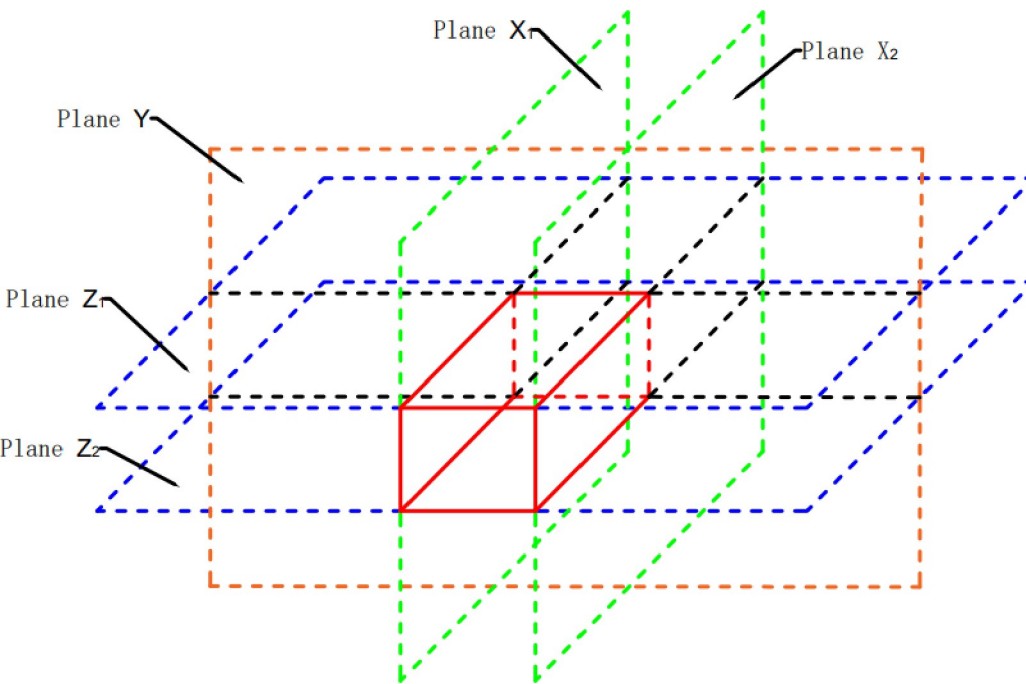

**Fig 9. Schematic diagram of the stratum virtual surfaces.**

1. When the internal roaming camera moves in the X direction, the stratum virtual surfaces Y, $Z_1$ and $Z_2$ do not need to update their positions and textures and control the stratum virtual surfaces $X_1$ and $X_2$ to follow the camera movement.

2. When the internal roaming moves along the Y direction, the stratum virtual plane $X_1$, $X_2$, $Z_1$ and $Z_2$ need not update the position and texture to control the stratum virtual plane Y to follow the camera movement.

3. When the internal roaming moves in the Z direction, the stratum virtual surfaces $X_1$, $X_2$ and $Y_2$ do not need to update their positions and textures and control the stratum virtual surfaces $Z_1$ and $Z_2$ to follow the camera movement.

The distance from the internal roaming camera (viewpoint) to the five stratum virtual planes $X_1$, $X_2$, Y, $Z_1$ and $Z_2$ is kept constant regardless of the camera movement.

## 2.3 Internal and external boundary identification method

The key to the combination of external and internal roaming of the stratum model is to determine the location of the roaming camera (viewpoint). The location of the roaming camera (viewpoint) is determined by analyzing the number of intersections between the shooting direction (line of sight) of the roaming camera (viewpoint) and the model. As shown in Fig 10, if there are an even number of intersections between the camera's shooting direction and the stratum model, the camera is outside the stratum model and is an external roaming camera, such as Camera 1; and if there are an odd number of intersections between the camera's shooting direction and the stratum model, the camera is inside the stratum model and is an internal roaming camera, such as Camera 2. It is worth noting that intersections 2 and 3 are the parts of the two stratum surfaces that overlap and are counted as two intersections each; such intersections are Class I intersections, while intersections 1, 4, 5 and 6 are the outer surfaces of the

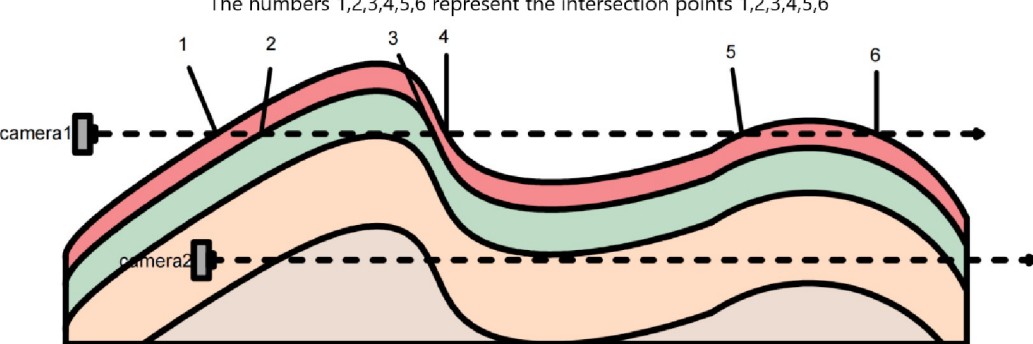

The numbers 1,2,3,4,5,6 represent the intersection points 1,2,3,4,5,6

**Fig 10. Schematic diagram of the shooting direction (line of sight) and stratum.**

entire stratum model without stratum overlap and are counted as only one intersection, which is a Class II intersection. Therefore, Camera 1 has eight intersections with the stratum model, and Camera 2 has seven intersections with the stratum model. The number of intersections is:

$$P = (P_\mathrm{I} \times 2) + P_\mathrm{II}$$

where P is the number of intersections, $P_\mathrm{I}$ is the number of Class I intersections and $P_\mathrm{II}$ is the number of Class II intersections.

## 3 Results and discussion

Based on the formation mechanism of the stratum virtual surface, and the inner and outer boundary identification method described above, actual tests of the 3D stratum internal and external roaming are carried out on actual projects to verify the method's feasibility.

### 3.1 Case overview

According to the detailed survey report of the case project, 85 boreholes were completed in the site area, with an average exploration depth of 26.96 m. The primary lithologies of the strata in the site area are prime fill, widely distributed in the site area, exposed in all the land boreholes. The layer thickness ranges from 0.7 m to 4.0 m, with an average thickness of 1.76 m. There is chalky clay widely distributed in the site area and exposed in all the boreholes. The layer thickness ranges from 0.5 to 5.5 m, with an average thickness of 3.57 m. Medium to coarse sand is distributed over most of the site area. The layer thickness ranges from 0.4 to 5.4 m, with an average thickness of 1.85 m. Based on the above survey reports, a three-dimensional stratum model was established based on the borehole data, as shown in Fig 11.

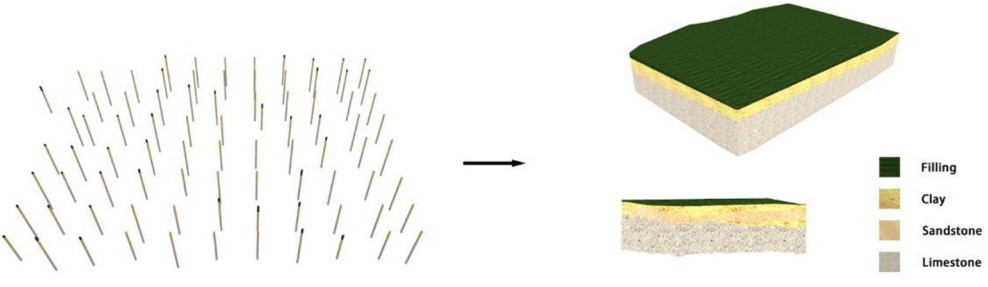

Filling

Clay

Sandstone

Limestone

**Fig 11. 3D stratum model.**

### 3.2 Case study: 3D stratum roaming analysis

According to the actual profile of the project, Based on application WebStorm,3D visualization techniques such as the 3D roaming engine(Three.js), jQuery framework and bootstrap framework were used to analyze the 3D roaming project.

1. Use the 3D roaming engine to define the essential elements of the scene, camera, light sources, renderers, controllers, etc.

2. According to the formation mechanism of the stratum virtual surface, use the 3D roaming engine to define the stencil test comparison function and draw the stencil mesh. As the stencil mesh is invisible, it is necessary to turn off the color writing of the material. After drawing and defining Plane $X_1$, Plane $X_2$, Plane Y, Plane $Z_1$ and Plane $Z_2$, set the comparison function to always pass the test, i.e., Fc (always), and open the first stencil test. After the test is complete, the outer surface stencil value is subtracted by one, the inner surface stencil value is incremented by one, and the Stencil mesh is rendered. The clipping mesh is drawn with Plane $X_1$, Plane $X_2$, Plane Y, Plane $Z_1$ and Plane $Z_2$ as the geometry model, the color writing of the material is switched on, the material is the lithological texture pattern corresponding to each stratum and the comparison function is set to pass the test when the stencil value is different, i.e., $F_c$ (NotEqual), and the second stencil test is switched on.

3. Use the rendering order value to define the rendering order. The stratum has four strata, each with five profiles to be drawn, and the rendering order value for the stencil mesh is

$$R = \begin{Bmatrix} R_1 \\ R_2 \\ R_3 \\ R_4 \end{Bmatrix} = \left\{ X_1 \quad X_2 \quad Y \quad Z_1 \quad Z_2 \right\} = \begin{Bmatrix} 1 & 2 & 3 & 4 & 5 \\ 6 & 7 & 8 & 9 & 10 \\ 11 & 12 & 13 & 14 & 15 \\ 16 & 17 & 18 & 19 & 20 \end{Bmatrix}$$

The rendering order values for the clipping mesh are

$$C = \begin{Bmatrix} C_1 \\ C_2 \\ C_3 \\ C_4 \end{Bmatrix} = \left\{ X_1 \quad X_2 \quad Y \quad Z_1 \quad Z_2 \right\} = \begin{Bmatrix} 1.1 & 2.1 & 3.1 & 4.1 & 5.1 \\ 6.1 & 7.1 & 8.1 & 9.1 & 10.1 \\ 11.1 & 12.1 & 13.1 & 14.1 & 15.1 \\ 16.1 & 17.1 & 18.1 & 19.1 & 20.1 \end{Bmatrix}$$

1. Control of the clipping mesh. Let the distances from the camera (viewpoint) to each of the clipping meshes, $X_1$, $X_2$, Y, $Z_1$ and $Z_2$, be

$$L = \left\{ X_1 \quad X_2 \quad Y \quad Z_1 \quad Z_2 \right\} = \left\{ 50 \quad 50 \quad 60 \quad 50 \quad 50 \right\}$$

The profile is then controlled in the same way as the profile control above.

1. Internal and external recognition. Define ray(origin, direction) to create a ray, with the origin being the camera's position and the direction being the direction of the camera shot. The number of intersection points between the ray and the stratum model determines the camera position.

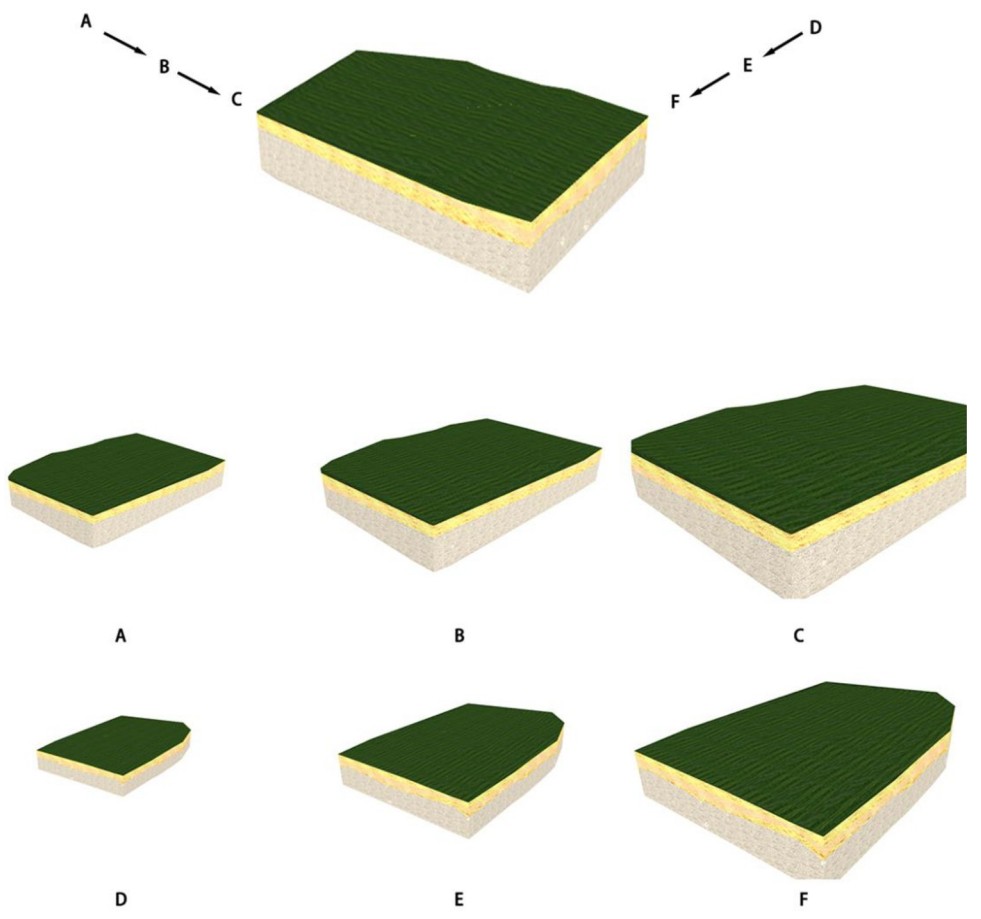

**Fig 12. 3D stratum external roaming effect.**

### 3.3 Case 3D stratum roaming effect

When the number of intersection points P is even, the camera is located outside the model. This is the external roaming camera, which cannot execute the profile complement function, normal rendering of the 3D stratum model, or use the first-person view controller in the 3D roaming engine, as shown in Fig 12, where viewpoints A, B, C and viewpoints D, E, F are gradually closer to the stratum model points.

When the number of intersection points P is odd, the camera is located inside the model, and this is the internal roaming camera, which executes the clipping surface cut-and-fill function and renders five clipping meshes, as shown in Fig 13

When the viewpoint enters from the outside of the model to the inside of the model and switches from external to internal roaming, the effect is shown in Fig 14, where viewpoint A and viewpoint B are the external points of the model, and viewpoint C is the internal point of the model.

### 3.4 Discussion

As seen from the 3D stratum roaming results in Section 3.3, the 3D stratum internal roaming technique based on internal and external boundary recognition shows significant advantages in many ways. First, the loading of the models and the implementation of the external roaming

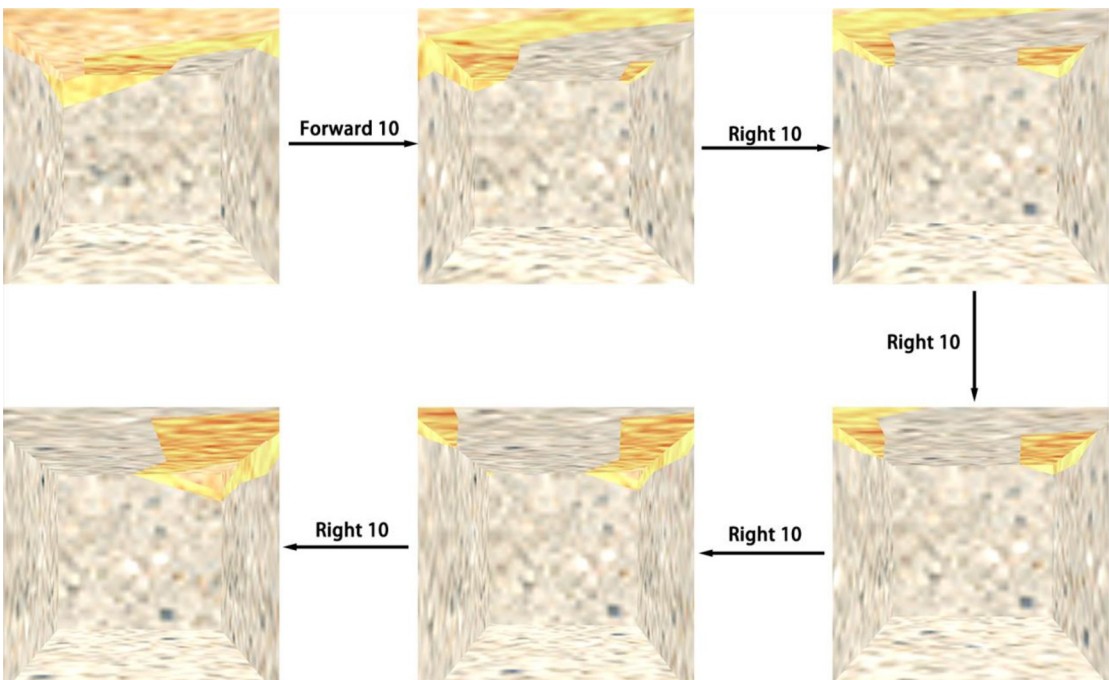

**Fig 13. 3D stratum internal roaming effect.**

of the 3D stratum models based on the 3D roaming engine (Three.js) has four main advantages:

1. Asynchronous loading. The 3D roaming engine supports a mechanism for loading models asynchronously. It can split the model data into smaller chunks and load them on demand without waiting for the entire model data to be fully loaded before rendering. This

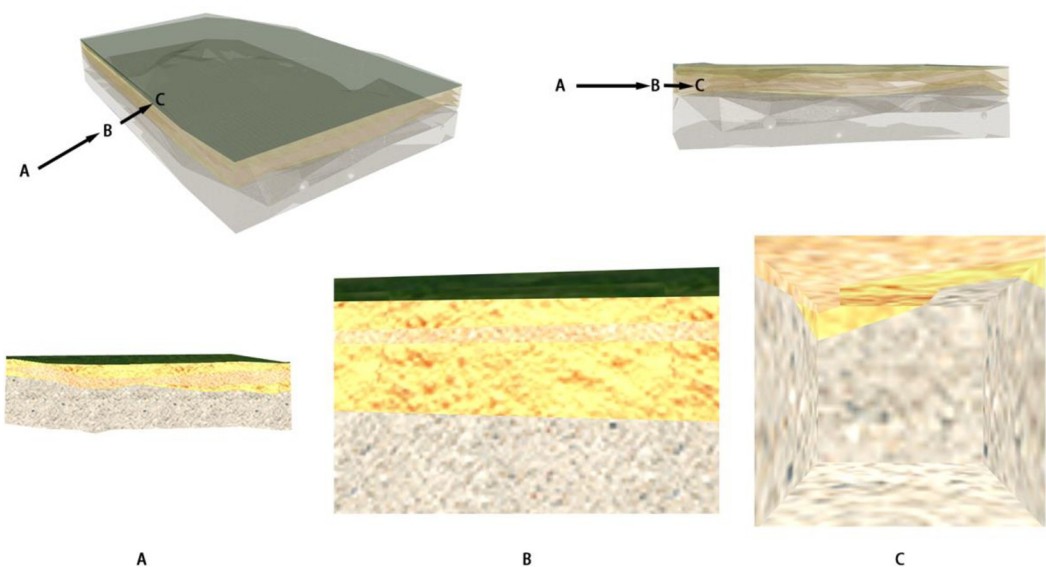

**Fig 14. Effect of switching from external roaming to internal roaming.**

asynchronous loading can significantly increase the loading speed, especially for large models and complex scenes, allowing users to view and interact with the models quickly.

2. Compression and Optimization. The 3D roaming engine offers the ability to compress and optimize the models. By compressing the model data, removing redundant information and using appropriate data structures, the file size of the model can be significantly reduced, resulting in faster network transmissions and loading times. During the loading process, the 3D roaming engine also performs cascading shader and level of detail (LOD) operations according to the actual requirements, reducing the pressure on the GPU and CPU, and increasing the rendering speed.

3. Caching mechanism. The 3D roaming engine has an internal caching tool that allows the loaded model data to be cached in the browser's local storage for subsequent access and use. In this way, when the user reaccesses the same model, it can be loaded directly from the local cache, reducing requests to the server, lessening the waiting time and increasing the loading speed.

4. Parallel loading. The 3D roaming engine supports the parallel loading of multiple models at the same time. By loading in parallel, faster loading can be achieved using multiple threads or multiple requests from the browser, taking full advantage of the multicore processing power of modern browsers. This allows more models to be loaded in the same amount of time, improving the overall loading efficiency.

Compared with other technologies which is difficult to realize the problem of internal roaming in 3D stratigraphic models, the technology can meet the needs of both internal and external roaming in engineering practice, effectively solving the problems faced by the internal roaming of the 3D stratum models and providing sound roaming effects. What's more, the technology uses a more concise way to switch between internal roaming and external roaming. When the number of intersection points between the ray based on the camera's line of sight and the entire 3D stratum model is odd, 3D external roaming is enabled. When the number of intersection points between the ray and the entire 3D stratum model is even, and when the number of intersection points between the ray and the entire 3D stratum model is. When the number of intersections between the ray and the 3D stratum model is even, the roaming is switched to internal roaming, making the entire process simpler and more efficient.

The loading efficiency of the model and the stratum virtual surface was improved to some extent by controlling the stratum virtual surface when performing the 3D stratum interior roaming. The experimental data are shown in Fig 15. Without movement control of the stratum plane, the loading response time was reduced to approximately 0.5 times when the viewpoint was moved in the X direction, approximately 0.3 times when the viewpoint was moved in the Y direction and approximately 0.5 times when the viewpoint was moved in the Z direction. This shows that the technique is able to significantly increase the loading speed, and thus reduces the user's response time when the viewpoint is moved in a single direction. However, when the viewpoint is moved in both the X and Y directions or along the Y and Z directions, the load response times are similar, and when the viewpoint is moved in both the X and Z directions, the load response time is reduced to approximately 0.8 times. It can be seen that the optimization of the loading rate is relatively weak when the viewpoint is moved in multiple directions.

The weighted average of the load response times based on all possible directions:

$$t_C = \frac{(0.5 + 0.5 + 0.33 + 1 + 0.8 + 1)t_u}{8} = 0.688 t_u.$$

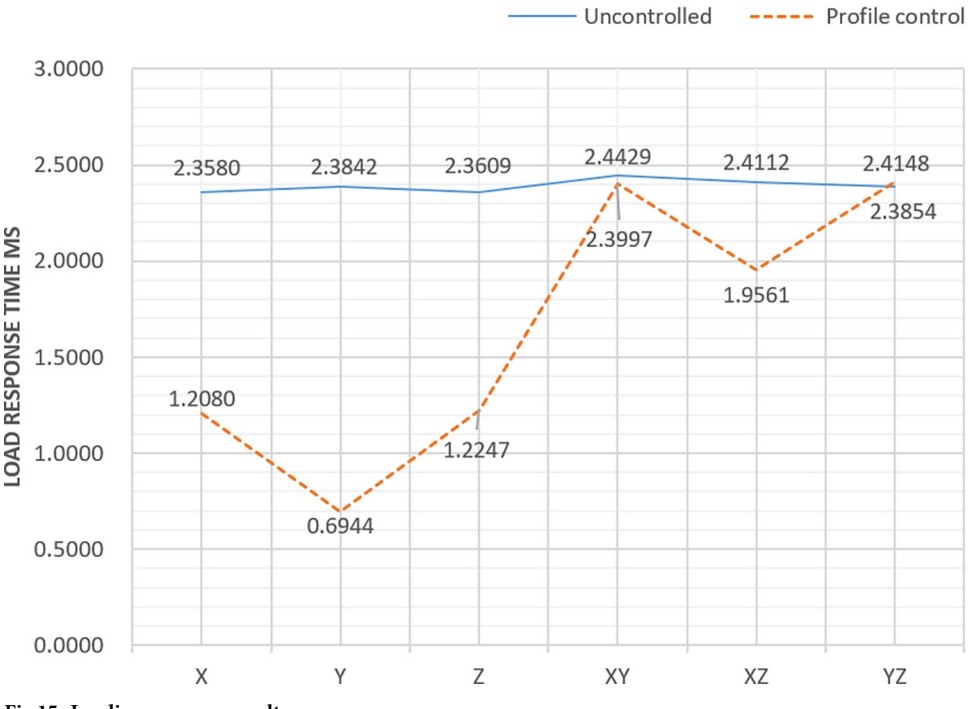

**Fig 15. Loading response results.**

$$v_C = \frac{v_u}{0.688} = 1.45 v_u$$

where $t_c$ is the average response time with virtual surface control, $t_u$ is the average response time without virtual surface control, $v_c$ is the average rate with virtual surface control, and $v_u$ is the average rate without virtual surface control. Although the optimization of the loading rate is relatively weak when the viewpoint is moved in multiple directions, the average rate with stratum virtual surface control, $v_c$, is still 1.45 times higher than the average rate without control, 1.45 times the uncontrolled average rate.

A technical solution that organically combines internal and external roaming conveys great convenience to engineering practices. This innovative technical solution offers new ideas and possibilities for research and practical applications in related fields.

## 4 Conclusion

1. This paper addresses the situation where the internal hollow of a 3D stratum model does not allow for internal roaming of the model, uses a 3D roaming engine to achieve internal roaming of a 3D stratum model using the stratum virtual surface method and combines 3D external roaming and internal roaming through the internal and external boundary identification method so that 3D stratum roaming is not limited to external or internal roaming. Compared to other methods,there is no need to change the scene during the switch between internal and external roaming, and there is no reliance on a tracker to position the camera.

2. Using a 3D roaming engine to load models and implement 3D roaming has the advantages of asynchronous loading, compression optimization, caching mechanisms, parallel loading, etc. Through the control of the stratum virtual surface, the loading efficiency of internal

roaming is optimized. Among them, the loading rate is increased by approximately two times when the viewpoint is moved along the X direction, nearly three times when it is moved along the Y direction, and approximately two times when it is moved along the Z direction; although the loading rate performs similarly when it is moved along multiple orders at the same time, the overall average loading rate is also increased by approximately 1.45 times.

3. This technology has certain limitations, that is, when roaming inside the 3D stratum, the stratum virtual plane will be limited to a certain surface (such as the five stratum virtual planes in this paper). Therefore, this problem can be overcome in the future to optimize the effect of 3D formation interior roaming.

4. The free roaming implementation method of the 3D stratum model based on internal and external boundary identification proposed in this paper can be applied not only to intelligent geology, geology and geotechnical engineering professions but also to mining and petroleum. In industries such as mining and oil, it brings great convenience to work by providing 3D visualization of underground roaming.

## Supporting information

**S1 Data. Stratum modeling data.**
(XLSX)

## Acknowledgments

Thanks to the corresponding authors for their guidance and to other authors for their contributions to this article.

## Author Contributions

**Conceptualization:** Yusen Zhong, Zhen Liu.

**Data curation:** Yusen Zhong.

**Formal analysis:** Yusen Zhong.

**Funding acquisition:** Cuiying Zhou.

**Investigation:** Yusen Zhong.

**Methodology:** Yusen Zhong, Cuiying Zhou.

**Project administration:** Yusen Zhong, Zhen Liu.

**Resources:** Zhen Liu, Cuiying Zhou.

**Software:** Yusen Zhong.

**Supervision:** Zhen Liu, Cuiying Zhou.

**Validation:** Yusen Zhong.

**Visualization:** Yusen Zhong.

**Writing – original draft:** Yusen Zhong.

**Writing – review & editing:** Zhen Liu.

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
