## [Decision Letter · Decision Letter 0]

15 Sep 2023

PONE-D-23-22040

Free roaming of 3D stratum models based on internal and external boundary identification

PLOS ONE

Dear Dr. Liu,

Thank you for submitting your manuscript to PLOS ONE. After careful consideration, we feel that it has merit but does not fully meet PLOS ONE’s publication criteria as it currently stands. Therefore, we invite you to submit a revised version of the manuscript that addresses the points raised during the review process.

We look forward to receiving your revised manuscript.

Kind regards,

Vijayalakshmi Kakulapati, Ph.D

Academic Editor

PLOS ONE

Journal Requirements:

   "The research is supported by the National Natural Science Foundation of China (NSFC)  (Grant Numbers: 42293354, 42293351, 42293355, 42277131, 41977230).

These NSFC fundings are all awarded by the author Cuiying Zhou.

The model software and equipment are funded by these sponsors."

5. Please amend the manuscript submission data (via Edit Submission) to include author Yusen Zhong.

6. Please ensure that you refer to Figure 10 in your text as, if accepted, production will need this reference to link the reader to the figure.

7. We note you have included a table to which you do not refer in the text of your manuscript. Please ensure that you refer to Table 1 in your text; if accepted, production will need this reference to link the reader to the Table.

Reviewers' comments:

Reviewer's Responses to Questions

**Comments to the Author**

1. Is the manuscript technically sound, and do the data support the conclusions?

Reviewer #1: Yes

Reviewer #2: Partly

2. Has the statistical analysis been performed appropriately and rigorously? 

Reviewer #1: Yes

Reviewer #2: No

3. Have the authors made all data underlying the findings in their manuscript fully available?

Reviewer #1: Yes

Reviewer #2: No

4. Is the manuscript presented in an intelligible fashion and written in standard English?

Reviewer #1: Yes

Reviewer #2: No

5. Review Comments to the Author

Reviewer #1: 1. General Comments:

In this paper, a 3D roaming engine is used to connect the interior and exterior of the 3D model. Based on the internal and external boundary identification method, the stratum virtual surface is used to realize the combination of external and internal roaming of the 3D stratum model, and the internal and external roaming of the 3D stratum model is established. The loading efficiency of internal roaming is optimized by controlling the stratum virtual surface. However, However, there are several issues with this manuscript. It should therefore be systematically revised. The following comments may help enhance the quality of this manuscript.

2. Detailed Comments:

1. It is recommended to label the meaning of the rectangle in Fig. 2, and Fig. 4 as well.

2. It is recommended that the dotted lines representing the planes in Figure 9 be of a different color to make the planes clearer.

3. A scale is advised to be added to the vertical coordinates of Table 1.

4. Numbers 1-6 in Figure 10 are supposed be annotated with their meaning in the figure.

5. List the inadequacies of current researches and the innovations of this article point by point in the introduction. Here, the reviewer recommends the following articles for the author's references:

Rapid Path Extraction and Three‐Dimensional Roaming of the Virtual Endonasal Endoscope

Three-dimensional coupled hydromechanical analysis of localized joint leakage in segmental tunnel linings

6. The notes in Figure 11 should be in English , not in another language.

Reviewer #2: 1. What are the real time applications of the proposed model. Verify your model with a real time application.

2. Discuss the novelty clearly in the abstract. How the devised model is performed better than the other techniques.

3. Write the motivation behind the study in the introduction part.

4. Write the paper structure at the end of the Introduction.

5. Improve the writing quality of the paper. The entire manuscript should be proofread throughly. The grammatical and typo errors should be corrected.

6. Related work section should be added. Also, discuss the research gaps of the existing methods.

7. Provide the numbers for the equations. Define all the variables in the equations.

8. Compare your model with other related methods.

9. Discuss the limitations of your model.

10. Discuss further extension of your model.

6. PLOS authors have the option to publish the peer review history of their article (what does this mean?). If published, this will include your full peer review and any attached files.

Reviewer #1: **Yes: **Chenjie Gong

Reviewer #2: No

---

## [Author Response · Author response to Decision Letter 0]

12 Oct 2023

Academic Editor, Concern # 1: Please ensure that your manuscript meets PLOS ONE's style requirements, including those for file naming.

Author response: Thank you for reminding me. We have modified the format of this article according to the template you provided.

Academic Editor, Concern # 2: Thank you for stating the following financial disclosure: 

 "The research is supported by the National Natural Science Foundation of China (NSFC) (Grant Numbers: 42293354, 42293351, 42293355, 42277131, 41977230).

These NSFC fundings are all awarded by the author Cuiying Zhou.

The model software and equipment are funded by these sponsors."

Author response: Thank you for reminding me. We have stated in the cover letter as follows” This research is supported by the National Natural Science Foundation of China (Grant Numbers: 42293354, 42293351, 42293355, 42277131, 41977230). These fundings are all awarded by the corresponding author Cuiying Zhou. We declare that the funder has no known competing financial interests or personal relationships that could have appeared to influence the work reported in this paper. The funders had no role in study design, data collection and analysis, decision to publish, or preparation of the manuscript.”

Academic Editor, Concern # 3: In your Data Availability statement, you have not specified where the minimal data set underlying the results described in your manuscript can be found. PLOS defines a study's minimal data set as the underlying data used to reach the conclusions drawn in the manuscript and any additional data required to replicate the reported study findings in their entirety. All PLOS journals require that the minimal data set be made fully available. For more information about our data policy, please see http://journals.plos.org/plosone/s/data-availability.

Author response: Thank you for reminding me. Relevant data supporting the results of this study have been uploaded to a file named " stratum modeling data".

Academic Editor, Concern # 4: We note that you have stated that you will provide repository information for your data at acceptance. Should your manuscript be accepted for publication, we will hold it until you provide the relevant accession numbers or DOIs necessary to access your data. If you wish to make changes to your Data Availability statement, please describe these changes in your cover letter and we will update your Data Availability statement to reflect the information you provide.

Author response: Thank you for reminding me. 

Relevant data supporting the results of this study have been uploaded to a file named " stratum modeling data".

Academic Editor, Concern # 5: Please amend the manuscript submission data (via Edit Submission) to include author Yusen Zhong.

Author response: Thank you for your reminder. We have revised the submission data.

Academic Editor, Concern #6: Please ensure that you refer to Figure 10 in your text as, if accepted, production will need this reference to link the reader to the figure.

Author response: Thank you for your reminder. We refer to Figure 10 on line 261 of this paper.

Academic Editor, Concern # 7: We note you have included a table to which you do not refer in the text of your manuscript. Please ensure that you refer to Table 1 in your text; if accepted, production will need this reference to link the reader to the Table.

Author response: Thank you for your reminder. We refer to Table 1 on line 397 of this paper.

Reviewer#1, Concern # 1: It is recommended to label the meaning of the rectangle in Fig. 2, and Fig. 4 as well.

Author response: Thank you for your advice. Figure 2 and Figure 4 have been modified and we have indicated the meaning of the rectangles marked in Figure 2 and Figure 4.

Reviewer#1, Concern # 2: It is recommended that the dotted lines representing the planes in Figure 9 be of a different color to make the planes clearer.

Author response: Thank you for your advice. Figure 9 has been modified and we have used dashed lines of different colors to represent the plane of Figure 9.

Reviewer#1, Concern # 3: A scale is advised to be added to the vertical coordinates of Table 1.

Author response: Thank you for your advice. We have added the scale in Table 1.

Reviewer#1, Concern # 4: Numbers 1-6 in Figure 10 are supposed be annotated with their meaning in the figure..

Author response: Thank you for your advice. We have commented on numbers 1 through 6 in Figure 10. 

Reviewer#1, Concern # 5: List the inadequacies of current researches and the innovations of this article point by point in the introduction. Here, the reviewer recommends the following articles for the author's references: Rapid Path Extraction and Three-Dimensional Roaming of the Virtual Endonasal Endoscope.Three-dimensional coupled hydromechanical analysis of localized joint leakage in segmental tunnel linings.

Author response: Thank you for your advice. First of all, we have listed the shortcomings of the current research and the innovations of this paper point by point in the introduction. Secondly, we have referred to the two articles you provided and quoted them in the article.

Reviewer#1, Concern # 6: The notes in Figure 11 should be in English , not in another language.

Author response: Thank you for reminding me and we have modified the annotations in Figure 11.

Reviewer#1, Concern # 7: Mention the crisp conclusions in the conclusion section. Use academic language to write conclusions.

Author response: Thank you for your professional comments on our article. We have rewritten the conclusion section according to your suggestion. The conclusion (1), and (2) show the contributions of this thesis and the conclusion (3) shows the limitations of this paper.

Reviewer#2, Concern # 1: What are the real time applications of the proposed model. Verify your model with a real time application.

Author response: Thank you for your question. This model is mainly applied to three-dimensional underground space roaming. Based on the 3D stratigraphic model, it is easy to analyze the internal structure of the 3D stratigraphic model by roaming inside the 3D stratigraphic model. We developed the model primarily through the 3D roaming engine (Three.js), using WebStorm. Among them, the 3D Roaming Engine (Three.js) is a toolkit for JavaScript. A brief introduction is provided on lines 98 through 112 of this paper. Based on your suggestion, a brief explanation is given in lines 296 through 299 of this paper.

Reviewer#2, Concern # 2: Discuss the novelty clearly in the abstract. How the devised model is performed better than the other techniques.

Author response: Most studies on 3D scene roaming do not consider the case of viewpoint entering into 3D model. Moreover, whether it is a hollow 3D model or a solid 3D model, as long as the viewpoint enters the interior of the 3D model, the internal structure of the model cannot be accurately displayed. However，In geotechnical and geological engineering, it is often necessary to carry out visual analysis of the interior of 3D formation models.And the technology can meet the needs of both internal and external roaming in engineering practice, effectively solving the problems faced by the internal roaming of the 3D stratum models and providing sound roaming effects. What's more, the technology uses a more concise way to switch between internal roaming and external roaming. This paper gives a brief analysis in lines 381 to 394.

Reviewer#2, Concern # 3: Write the motivation behind the study in the introduction part.

Author response: Thank you for your advice. The research motivation is added in lines 93 to 97 of this paper.

Reviewer#2, Concern # 4: Write the paper structure at the end of the Introduction.

Author response: Thank you for your advice. We have added paper structure at the end of the introduction

Reviewer#2, Concern # 5: Improve the writing quality of the paper. The entire manuscript should be 

proofread throughly. The grammatical and typo errors should be corrected.

Author response: Thank you for reminding me. We have checked the full text and made corrections where there are errors.

Reviewer#2, Concern # 6: A Related work section should be added. Also, discuss the research gaps of the existing methods.

Author response: Thank you for your advice. In lines 93 to 97 of the introduction, we add an analysis of existing research gaps.

Reviewer#2, Concern # 7:: Provide the numbers for the equations. Define all the variables in the equations.

Author response: Thank you for your advice. We've defined all the variables in the equation.

Reviewer#2, Concern # 8: Compare your model with other related methods.

Author response: Thank you for pointing this out. In lines 436-438, this method is compared with existing methods and their advantages are explained.

Reviewer#2, Concern # 9: Discuss the limitations of your model.

Author response: Thank you for pointing this out. The limitations of this method are explained in lines 450-454 of this paper.

Reviewer#2, Concern # 10: Discuss further extension of your model.

Author response: Thank you for pointing this out. At the end of this paper, the further extension of this model is discussed.

---

## [Decision Letter · Decision Letter 1]

6 Mar 2024

Free roaming of 3D stratum models based on internal and external boundary identification

PONE-D-23-22040R1

Dear Dr. Liu,

We’re pleased to inform you that your manuscript has been judged scientifically suitable for publication and will be formally accepted for publication once it meets all outstanding technical requirements.

Kind regards,

Vijayalakshmi Kakulapati, Ph.D

Academic Editor

PLOS ONE

All comments are addressed

**Comments to the Author**

Reviewer #1: All comments have been addressed

Reviewer #3: All comments have been addressed

---

## [Editor Report · Acceptance letter]

15 Mar 2024

PONE-D-23-22040R1 

PLOS ONE

Dear Dr. Liu, 

I'm pleased to inform you that your manuscript has been deemed suitable for publication in PLOS ONE. Congratulations! Your manuscript is now being handed over to our production team.

Kind regards, 

on behalf of

Dr. Vijayalakshmi Kakulapati 

Academic Editor

PLOS ONE